# Multi-Objective Design for Critical Supporting Parameters of Vacuum-Insulated Glazing with a Case Study

Yifu Zhang [1],*, Wei Yuan [1], Lianjie Han [1], Ruihong Zhang [1,2] and Xiaobo Xi [1,2]

[1] School of Mechanical Engineering, Yangzhou University, Yangzhou 225127, China; mx120210538@yzu.edu.cn (W.Y.); mx120210537@stu.yzu.edu.cn (L.H.); zhangrh@yzu.edu.cn (R.Z.); xbxi@yzu.edu.cn (X.X.)

[2] Jiangsu Engineering Center for Modern Agricultural Machinery and Agronomy Technology, Yangzhou 225127, China

* Correspondence: zyfu@yzu.edu.cn

**Abstract:** Vacuum-insulated glazing (VIG) has excellent sound and heat insulation and anti-fogging properties, having been a typical structure–function-integrated glass deep processing product. However, overlapping, vacancy and excessive spacing distance of the supporting pillars will increase the concentrated stress for the glass substrate, raising the potential risk of failure. Therefore, this study, aiming to address the high cost of sample preparation and the multiple factors affecting stress distribution, developed a multi-objective design for supporting stress. In this paper, a multi-objective optimization model was designed based on comprehensive mechanical analysis under square-distributed supporting. The critical supporting pillars radius as well as the critical spacing distance were solved and met the strength requirement. Case simulation demonstrated that a 0.2 mm or more radius and a 63 mm or less spacing distance for the supporting pillars were acceptable placement methods which conformed to the design requirements. This research will act as a theoretical reference for future studies, promoting the in-depth development of VIG and exploration of high-strength safety products.

**Keywords:** vacuum-insulated glazing; supporting pillar; multi-objective design; supporting spacing

## 1. Introduction

Currently, the global demand for carbon neutrality and energy utilization efficiency is increasing, and the energy crisis has attracted wide attention [1,2]. Emission reduction, substitution and energy saving are the main ways to achieve carbon peak, carbon neutrality and reduce energy consumption. Developing efficient insulated materials is an important for energy conservation. Excellent insulation materials can reduce energy loss and cost, producing great social benefits [3]. Vacuum insulation technology utilizes vacuum to eliminate convection and to achieve heat preservation and insulation, which has been applied in the petrochemical industry, refrigeration equipment, nuclear power and other fields [4–6].

Vacuum-insulated glazing (VIG) refers to a glass product that is separated by two flat glasses with supporting pillars, and the periphery is sealed to form an evacuated gap [7,8]. Due to the high inner vacuum degree, VIG can effectively block sound and heat transfer, and has excellent insulating, anti-fogging and anti-frosting properties. As a top-grade transparent insulation material, VIG has become important in glass deep processing around the world and has shown great market prospects in construction, facility agriculture, solar photovoltaic and automobile industry, etc. [9–11].

Since 1913, when Zoller A. [12] first conceptualized VIG and applied for the first patent, VIG has been theoretically studied. In 1991, the first sample was created at the University of Sydney [13–15]. Domestic research for VIG started in 2005, with high temperature side sealing technology created for its manufacturing [16]. In recent years, the research on VIG

mainly involves glass substrates, supporting pillar materials, edge sealing processes and thermal coupling mechanisms [17,18]. The optical and thermal parameters of glass panes have a crucial impact on the performance of insulation and transmittance for VIG. Float glass was applied as a main material for vacuum glass panes due to its good transparency, flatness and mechanical strength. In addition, the safety and photothermal performances were also improved by tempered, laminated or coating glass [19–21]. Supporting pillars were designed to prevent the deformation failure between two glass substrates under atmospheric pressure, among which metal and glass pillars were often used. Zhao J. et al. [22] executed screen printing to layout the supporting pillars for improving the thickness uniformity and reliability. In addition, the sealing materials with comprehensive features such as environmental protection, high stability and low sealing temperature are conducive to improve the mechanical property of VIG [23–25]. Laser brazing sealing process within a vacuum chamber was recently explored to optimize microstructure characteristics for VIG [26,27].

The placement for supporting pillars is of great importance during the VIG production. Overlapping, vacancy and excessive spacing of supporting pillars will increase the concentrated stress for glass plates, raising the potential risk of failure [28]. Therefore, under the continuous stress condition of VIG, this study carried out multi-objective optimization for supporting stress in view of the various factors affecting the stress distribution and the high cost of sample preparation. In this paper, a static model for glass pane was established under atmospheric pressure and the square-distributed supporting pillars; multi-objective optimization was designed based on bending stress and deflection to solve the optimal spacing and size parameters for the supporting pillars; and maximum deformation and stress were compared from an existing case to verify the feasibility using computer simulation.

The objective of this study was to introduce a multi-objective optimization for supporting pillars to improve the mechanical stability of VIG. This research will act as a theoretical reference for future studies, promoting the in-depth development of VIG and exploration of high-strength safety products.

## 2. Construction and Basic Assumptions

VIG is composed of two pieces of flat glass, sealing solder and supporting pillars (See Figure 1), wherein supporting pillars are arranged in a certain sequence to separate the two glass substrates. In this study, the supporting pillars were distributed in a square and equal space. Due to symmetry, only a single glass flat was discussed. Continuous stress and deformation of the glass panes occurred under atmospheric pressure and the assumptions for supporting characteristics were made as follows.

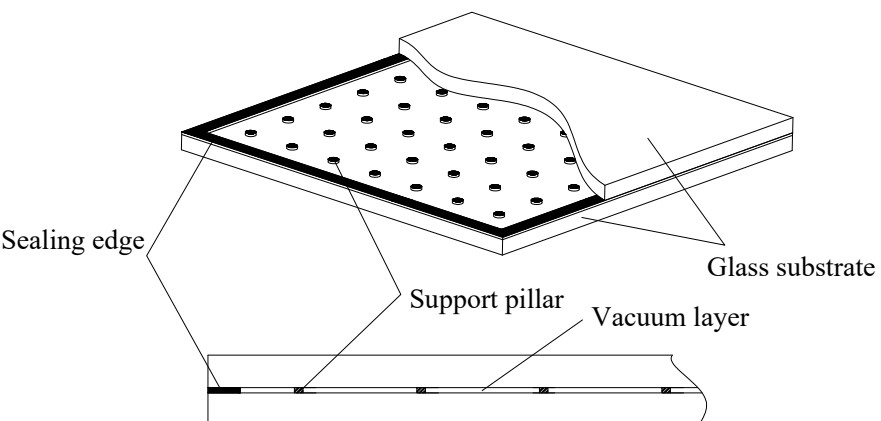

**Figure 1.** Structural diagram of vacuum-insulated glazing.

(1) Elastomer assumption: when the load on the glass does not exceed the limit, it exhibits elastic properties;

(2) Rigid supporting: the elastic modulus of supporting pillars is much larger than that of glass pane, and the glass deformation at the contact position is much larger than that of the supporting pillars. It can be assumed that the supporting pillars are regarded as rigid supports without deformation;

(3) Negligible edge displacement: the edge is designed as a fixed constraint after sealing, so that the edge has negligible displacement.

## 3. Mechanical Model Establishment

### 3.1. Stress Analysis

In order to avoid generating excessive stress and deformation within the two glass substrates, a series of supporting pillars are arranged inside the vacuum gap. However, the interaction between the glass pane and supporting pillars inevitably causes obvious stresses that cannot be ignored at several locations, mainly including three parts:

(1) The bending stress of the glass pane;

(2) The contact stress in the inner surface of the glass pane at the contact part with supporting pillars;

(3) The stress of the edge sealing part of the VIG.

The above stress is related to multifarious factors such as pillar material, pillar spacing and edge sealing. By rationally optimizing these parameters, these stresses could be controlled within the allowable range of material strength.

### 3.2. Bending Stress of the Glass Substrate

The pressure exerted by atmospheric pressure on the glass is a uniform load, and the supporting pillars are arranged regularly. Among them, extreme values are generated in the upper (outer) pane surface at the supporting point and the lower (internal) pane surface at the midpoint of the diagonal connection between the two supporting points.

A supporting pillar, except that near the outermost row, was taken as the center, and a square element at an equal distance from the adjacent supporting pillar was taken out (Figure 2). The stress characteristics were deduced according to the thin-plate theory, because the side length of the square element (i.e., spacing between pillars) was 7 times larger than the glass thickness. Under the uniform load $q_0$ from atmospheric pressure and supporting force $F$ from the supporting pillar, the maximum superimposed stress occurs on the upper surface of the glass at the supporting point:

$$f_{1,\max} = \frac{3q_0 a^2}{2\pi h^2}\left[(1+v)\ln\frac{2a}{\sqrt{1.6r^2+h^2}-0.675h}+\beta_2\right]-\beta_1\frac{q_0 a^2}{h^2} \tag{1}$$

where $f_{1,\max}$ is the maximum superimposed stress (MPa); $q_0$ is atmospheric pressure (MPa); $a$ is the pillar separation of the square unit (mm); $h$ is thickness of the glass substrate (mm); $v$ is the Poisson ratio; $r$ is the supporting pillar radius (mm); and $\beta_1$ and $\beta_2$ are structural coefficients.

Another square unit was still selected, the four corners were supported by four supporting pillars. The sides were free and the rotation and displacement were not constrained. The mechanical model could be simplified as a square plate supported by four corners subjected to uniform load, and the maximum bending stress occurred in the center of the glass pane. The calculation formula is as follows:

$$f_{2,\max} = \frac{\alpha_1 q_0 a^2}{h^2} \tag{2}$$

where $f_{2,\max}$ is the maximum bending stress (MPa) and $\alpha_1$ is the structural coefficient.

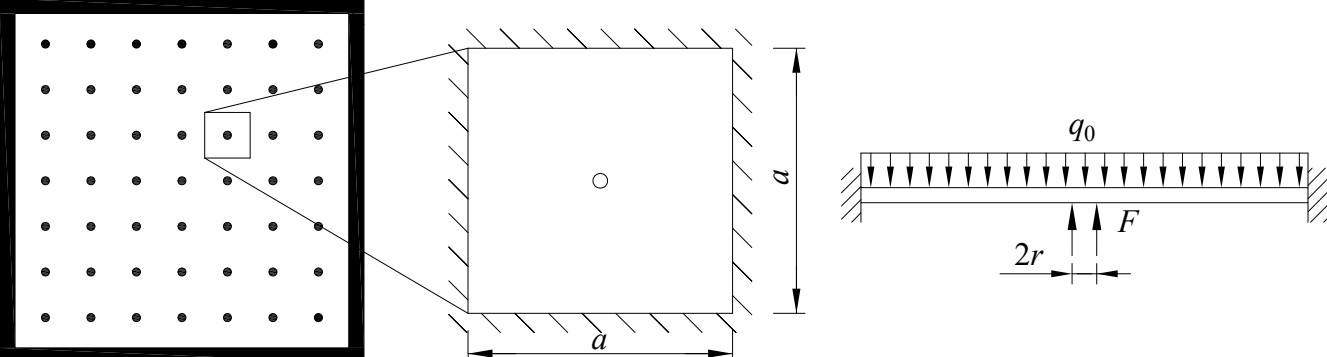

**Figure 2.** Simplified square unit and its force diagram. Where *a* is the pillar separation of the square unit, $q_0$ is the atmospheric pressure, *r* is the supporting pillar radius and *F* is the support force from the supporting pillar.

### 3.3. Contact Stress between the Glass Substrate and Supporting Pillars

Glass pane tends to create a stress concentration near the contact position with the supporting pillar. If the concentrated stress is too high, the excessive stress will lead to cracks on the pane surface. The supporting pillars are cylindrical and the diameter is much smaller than the thickness of glass pane. Therefore, the glass pane can be regarded as a semi-infinite space body and the pillar can be regarded as a flat-bottomed cylindrical indenter pressed into the glass. Due to the low tensile strength of the pane material, the primary cause of the crack is the maximum tensile stress on the glass surface:

$$f_{3,\max} = (-0.1373r^2 + 0.2862r + 0.0236)\frac{q_0 a^2}{\pi r^2} \tag{3}$$

where $f_{3,\max}$ is the maximum contact tensile stress (MPa).

### 3.4. Stress in the Edge Sealing Part of VIG

For the edge region, the atmospheric pressure taken by the glass is shared by the first row of supporting pillars and the edge sealing part, which is the direct reason of tensile stress. In this study, the glass unit corresponding to a row of supporting pillars was selected (Figure 3). Due to the symmetry, the supporting pillars within glass unit were located at the midpoint in the width direction, the width was *b* and the number of supporting pillars was *n*.

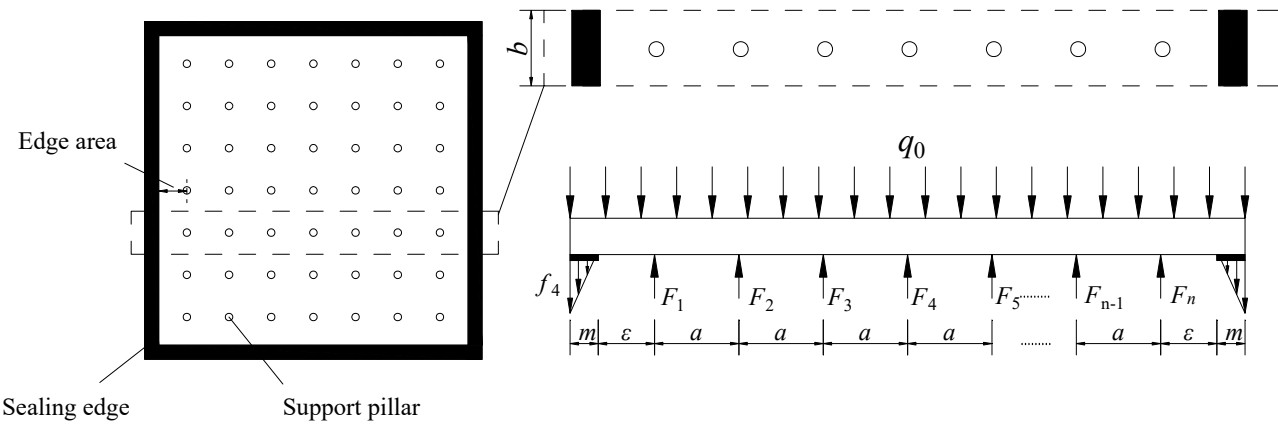

**Figure 3.** Schematic diagram of the force unit, the width of force unit is *b*, and supporting pillars are located at the midpoint in the width direction.

The upper surface of the glass is subjected to the uniform atmospheric pressure and the lower surface of the glass is subjected to the supporting force of the supporting pillars at

equal intervals. At the edge sealing part, the glass is subjected to tensile stress by the sealing material. In this study, the tensile stress was assumed to be a linear gradient distribution, and the largest value ($f_4$) was located in the outer edge. The inner edge of the edge sealing part was selected as the origin, and the force moment at this point was deduced. Each supporting pillar and atmospheric pressure contribute half of the force moment to the long and short sides.

The moment of force produced by atmospheric pressure on the right of origin is

$$2M_1 = \frac{q_0 \varepsilon^2 b}{2} + \frac{q_0(2\varepsilon+a)ab}{2} + \frac{q_0(2\varepsilon+3a)ab}{2} + \frac{q_0(2\varepsilon+5a)ab}{2}$$
$$+ \cdots + \frac{q_0[2\varepsilon+(2n-3)a]ab}{2} + \frac{q_0[\varepsilon+(n-1)a+\frac{1}{4}a]ab}{2} \tag{4}$$

where $\varepsilon$ is distance between the first supporting pillar and the inner sealing edge (mm); $b$ is width of the force moment unit (mm); and $n$ is the number of supporting pillars in the force moment unit.

The force moment produced by the supporting pillars on the right side of the origin is

$$2M_2 = F_1\varepsilon + F_2(\varepsilon + a) + F_3(\varepsilon + 2a) + \cdots + F_n[\varepsilon + (n-1)a] \tag{5}$$

$$F_2 = \ldots = F_n = q_0 ab \tag{6}$$

The force moment to the left of the origin is

$$M_l = \frac{f_4 m^2 b}{3} + \frac{q_0 m^2 b}{2} \tag{7}$$

where $f_4$ is the maximum line tension of the edge sealing part (MPa) and $m$ is the edge sealing width of VIG (mm).

According to the torque balance ($M_l = M_1 - M_2$),

$$f_4 = \left(\frac{q_0 \varepsilon^2}{2} + \frac{q_0 \varepsilon a}{2} + \frac{q_0 a^2}{8} - F_1\varepsilon - q_0 m^2\right)\frac{3}{2m^2} \tag{8}$$

The mathematic model showed that the maximum line tension of the edge sealing part ($f_4$) is mainly affected by the supporting force from the outermost supporting pillar ($F_1$) and the structural parameters ($m$ and $\varepsilon$). Obviously, the smaller the $F_1$, the greater the $f_4$. Therefore, the limit condition was considered, that is, when $F_1 = 0$, $f_4$ is the maximal value.

$$f_{4,\text{max}} = \left(\frac{\varepsilon^2}{2} + \frac{\varepsilon a}{2} + \frac{a^2}{8} - m^2\right)\frac{3q_0}{2m^2} \tag{9}$$

*3.5. Deformation of VIG*

The bending deformation of the glass pane in VIG is generated under permanent atmospheric pressure. Moreover, the contact deformation also occurs at the contact position between the supporting pillars and glass pane. The maximum bending deformation of the glass pane is generated at the midpoint of the simplified element (i.e., Figure 4), namely, the midpoint of the connection between two diagonal supporting pillars [16]:

$$w_1 = 0.32(1 - v^2)\frac{q_0 a^4}{E h^3} \tag{10}$$

where $w_1$ is the maximum bending deformation of the glass substrate (mm) and $E$ is the elastic modulus of the glass substrate (MPa).

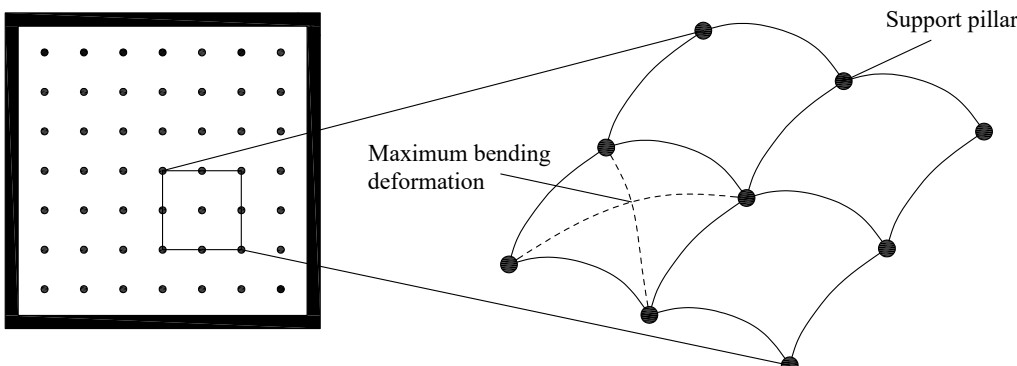

**Figure 4.** The maximum bending deformation of the glass substrate.

Moreover, the pressure from the supporting pillars on the glass pane leads to obvious compression deformation (Figure 5). The deformation is calculated according to the elastic contact mechanics theory [29]:

$$w_2 = 4(1 - \nu^2)\frac{q_0 a^2}{\pi^2 E} \tag{11}$$

where $w_2$ is the compression deformation (mm).

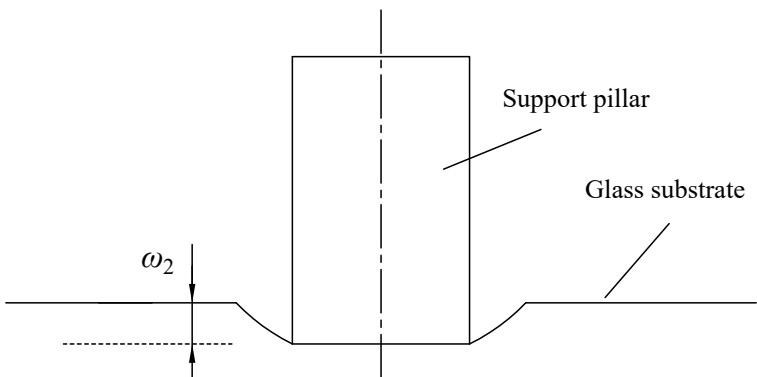

**Figure 5.** Contact deformation of the support pressed into the glass substrate.

## 4. Multi-Objective Optimization Design

The main research direction of parameter optimization is to design reasonable placement parameters for supporting pillars to minimize the stress and deformation of glass panes under atmospheric pressure. The main factors that affect stress distribution are supporting pillar spacing, supporting pillar size and the edge distance between the transition areas. In this study, the optimal objective for supporting pillar placement was primarily to minimize the supporting pillar number (i.e., the spacing) and radius with stable VIG mechanical and light-admitting property. This solution process with multiple optimization indicators is a multi-objective optimization problem, and the mathematical model can be expressed as:

$$\min f(X), X \in Z^*$$
$$s.t. \begin{cases} g_j(X) \le 0, (j = 1, 2, \ldots) \\ h_k(X) = 0, (k = 1, 2, \ldots) \end{cases} \tag{12}$$

where $f(X)$ is the objective function with the minimized goal; $X$ is the variable; $Z^*$ is the set of positive integers; $g_j(X)$ is the inequality constraint function; and $h_k(X)$ is the equality constraint function.

The optimization steps are mainly divided into variable design, objective function establishment, constraint condition analysis and model solution.

*4.1. Objective Function*

Within a square VIG, let *L* be the plate dimension and *S* the number of supporting pillars in each row. Then, the following equation is satisfied:

$$(S - 1)a + 2\varepsilon + 2m = L \tag{13}$$

Referring to the design requirements for the supporting pillar placement, objective functions were established as follows:

The number of supporting pillars within a single row was determined by

$$f_1 = \frac{L - 2m - 2\varepsilon}{a} + 1 \tag{14}$$

The supporting pillar radius was

$$f_2 = r \tag{15}$$

Therefore, the objective function in this study was $f(X)_{\min} = [f_1(X), f_2(X)]$.

*4.2. Variable Analysis*

The parameters of the supporting pillar placement mainly included supporting pillar spacing *a*, supporting pillar radius *r* and edge spacing *ε*. For a given glass material, the glass thickness *h* and Poisson ratio *ν* were not considered as variables. Thus, the variables were determined as $X = [x(1), x(2), x(3)] = (a, r, \varepsilon)$.

*4.3. Objective Function*

To acquire a reasonable spacing of supporting pillar placement, the stress limit should be considered. When the stress generated by the atmospheric pressure exceeds this limit, the glass pane is degraded.

The tensile stress distributed inside the glass pane is permanent, which should not exceed the glass strength.

$$f_{1,\max} \leq [\sigma_1], f_{2,\max} \leq [\sigma_1] \tag{16}$$

where $[\sigma_1]$ is the strength of the glass substrate under permanent stress, MPa.

The contact stress between the glass substrate and supporting pillars should avoid generating glass cracks, that is, the maximum contact stress ($f_{3,\max}$) should not exceed the local strength:

$$f_{3,\max} \leq [\sigma_2] \tag{17}$$

where $[\sigma_2]$ is the local strength of the glass substrate, MPa.

The tensile stress of the edge sealing part should not exceed the sealing strength and the marginal distance should not exceed the distance between the supporting pillars.

$$f_{4,\max} \leq [\sigma_3], \varepsilon \leq a \tag{18}$$

where $[\sigma_3]$ is the strength of the edge sealing area, MPa.

The total deformation of the glass pane cannot exceed the initial thickness of the vacuum layer (i.e., the height of supporting pillar):

$$2w_1 + 2w_2 \leq h_2 \tag{19}$$

where $h_2$ is the height of the supporting pillar (mm).

*4.4. Solution for Objective Function*

The purpose of multi-objective optimization is to solve the critical value of the supporting pillar placement parameters. In this study, the reported literature was selected to complete the multi-objective model solution. Due to the single-factor optimization design,

the aspect ratio of the glass pane was controlled to be 1, i.e., a/b = 1.0. The specific structure parameters are shown in Tables 1 and 2. See references [16,30] for further details.

**Table 1.** Coefficients when rectangular flat-plate functioned by uniform load [16].

| a/b | 1.0 | 1.2 | 1.4 | 1.6 | 1.8 | 2.0 |
|---|---|---|---|---|---|---|
| $\beta_1$ | 0.1386 | 0.1794 | 0.2094 | 0.2286 | 0.2406 | 0.2472 |
| $\beta_2$ | $-0.238$ | $-0.078$ | 0.011 | 0.053 | 0.068 | 0.067 |

**Table 2.** Structure parameters of VIG from reference [30].

| Structure Parameters | Abbreviations | Value | Unit |
|---|---|---|---|
| VIG size | $L \times L$ | $500 \times 500$ | mm $\times$ mm |
| Glass substrate thickness | $h$ | 5 | mm |
| Vacuum gap/pillar height | $h_2$ | 0.3 | mm |
| Edge sealing width | $m$ | 10 | mm |
| Atmosphere pressure | $q_0$ | 0.1 | MPa |
| Constant coefficient | $\alpha_1$ | 0.8719 | |
| | $\beta_1$ | 0.1386 | |
| | $\beta_2$ | $-0.238$ | |
| Glass density | | 2500 | kg m$^{-3}$ |
| Glass elastic modulus | | 72 | GPa |
| Glass Poisson ratio | $\nu$ | 0.24 | |

The fmincon function in the optimization toolbox of MATLAB software was adopted to solve the minimal value multi-objective optimization problem, and the format was:

$$[x,\text{fval}] = \text{fmincon}(@\text{objfun},x0,A,b,Aeq,beq,lb,ub)$$

where $x$ is the optimal solution of objective function; fval is the value of objective function at the optimal solution; @objfun is the function file name that called the objective function, in this paper, @objfun = @(x)fun(x(1),x(2),x(3)) = $(a, r, \varepsilon)$; x0 is the initial vector; A and b are the coefficient matrix and constant vector for inequality constraints, respectively; Aeq, and beq are the coefficient matrix and constant vector for equality constraints, respectively; lb and ub are the lower and upper limits of the variables, respectively.

In this paper, according to the reported material parameters in the literature [26], the MATLAB solution for the multi-objective critical model was completed, and the critical values for the radius and spacing distance of the supporting pillar were obtained as r = 0.1941 and a = 63.2455. Considering the safety and stability of VIG, the above results were rounded and the critical radius of the supporting pillar was 0.2 mm, while the critical spacing distance was 63 mm.

## 5. Case Simulation and Verification

### 5.1. Experimental Design

In this session, stress simulation for VIG was carried out to compare the stress and deformation distribution under different supporting pillar placement parameters. Glass substrate, with a 500 mm length, 500 mm width and 5 mm height, was selected to fabricate VIG. The supporting pillars with 0.3 mm height and 0.6 mm diameter were equally spaced between the glass panes. The width of the edge sealing was 10 mm.

The mechanical properties and parameters of the glass pane were designed based on the National Standard of Safety Glass for Construction (GB 15763.2-2005) [31], with a 2500 kg m$^{-3}$ density, a 72 GPa elastic modulus and a 0.24 Poisson ratio, consistent with Table 2. The supporting pillar was made of stainless steel with a density of 7800 kg m$^{-3}$, an elastic modulus of 200 GPa and a Poisson ratio of 0.3.

ANSYS workbench simulation was performed with different supporting pillar radii and spacing treatments. Supporting pillars were distributed in a square array, and three treatments were designed as follows: R3a63, with a 0.3 mm supporting pillar radius and 63 mm spacing; R3a50, with a 0.3 mm radius and 50 mm spacing; and R2a63, with a 0.2 mm radius and 63 mm spacing (Table 3).

**Table 3.** Experimental design for ANSYS simulation.

| Treatments | Supporting Pillar Radius (mm) | Supporting Pillar Spacing (mm) |
|---|---|---|
| R3a63 | 0.3 | 63 |
| R3a50 "[1]" | 0.3 | 50 |
| R2a63 | 0.2 | 63 |

[1] Supporting pillar radius and spacing were consistent with reference [30].

### 5.2. Simulation Method

First, a static structural module was established, the material properties for the glass pane and supporting pillars were set in Engineering Date. Then, we imported the VIG model and performed mesh division on the model. The mesh division of the glass pane and supporting pillars was refined according to the grid size, and the grid size of the glass pane was set to 2.5 mm while the supporting pillar was set to 0.15 mm. Finally, a uniform load of standard atmospheric pressure (0.1 MPa) was applied to the glass panel, and boundary was fixed to calculate the simulation results.

### 5.3. Results and Analysis

Table 4 compared the stress and deformation characteristics of the glass pane under different supporting pillar radii and spacing distances. In general, under permanent stress from atmospheric pressure and the supporting pillars, the deformation concentration point of the glass pane was located at the center of the square region, generated by the adjacent supporting pillars. Specifically, the largest deformation value occurred between the outermost row and second row of the supporting pillars. Simultaneously, the maximum stress point on the inner surface of the glass pane appeared at the position of the supporting pillar. These trends were consistent with previous research [26].

According to the National Standard of China "Safety glazing materials in building—Part 2: Tempered glass" (GB15763.2–2005), the concentrated load formed on the glass surface was suggested to be below 90 MPa. The ANSYS simulation showed that the stress peaks under the three treatments had a trend of 90 MPa > R2a63 > R3a63 > R3a50, which indicated that the VIG structures under R2a63, R3a63 and R3a50 treatments met the design requirements.

In order to ensure the mechanical stability of the VIG products, larger supporting pillar radii and smaller spacing distances could obtain better mechanical properties. In this study, a multi-objective solution was conducted based on the published literature [26], and a critical spacing distance for the supporting pillars was solved, i.e., R2a63 treatment. The stress peak of the glass pane under the R2a63 treatment, under a 0.2 mm supporting pillar radius and a 63 mm spacing distance was close to the recommended value (90 MPa). This demonstrated that the radius and spacing distance should be 0.2 mm or more and 63 mm or less, respectively, under this case, which verified the feasibility of the multi-objective solution with the critical load value as the constraint condition.

Li et al. [32] found that the critical supporting spacing for VIG was 70 mm when the glass substrate thickness was 5 mm and the supporting radius of the supporting pillar was 0.15 mm (guaranteeing the mechanical stability with the minimum supporting pillar number). However, the simulated results of ANSYS, using the above parameters, showed that the maximum stress on the inner surface was 109.05 MPa, which had exceeded the recommended value of national standard. The reason may be that the tensile stress inside the glass was not comprehensively considered a constraint condition. Meanwhile, supporting form such as spherical supporting, as well as the placement way also impacted stress concentration.

**Table 4.** The supporting stress and deformation of the glass substrate under different supporting pillar radii and spacing distances. R3a63, with a 0.3 mm supporting pillar radius and 63 mm spacing distance; R3a50, with a 0.3 mm supporting pillar radius and 50 mm spacing distance; R2a63, with a 0.2 mm supporting pillar radius and 63 mm spacing distance.

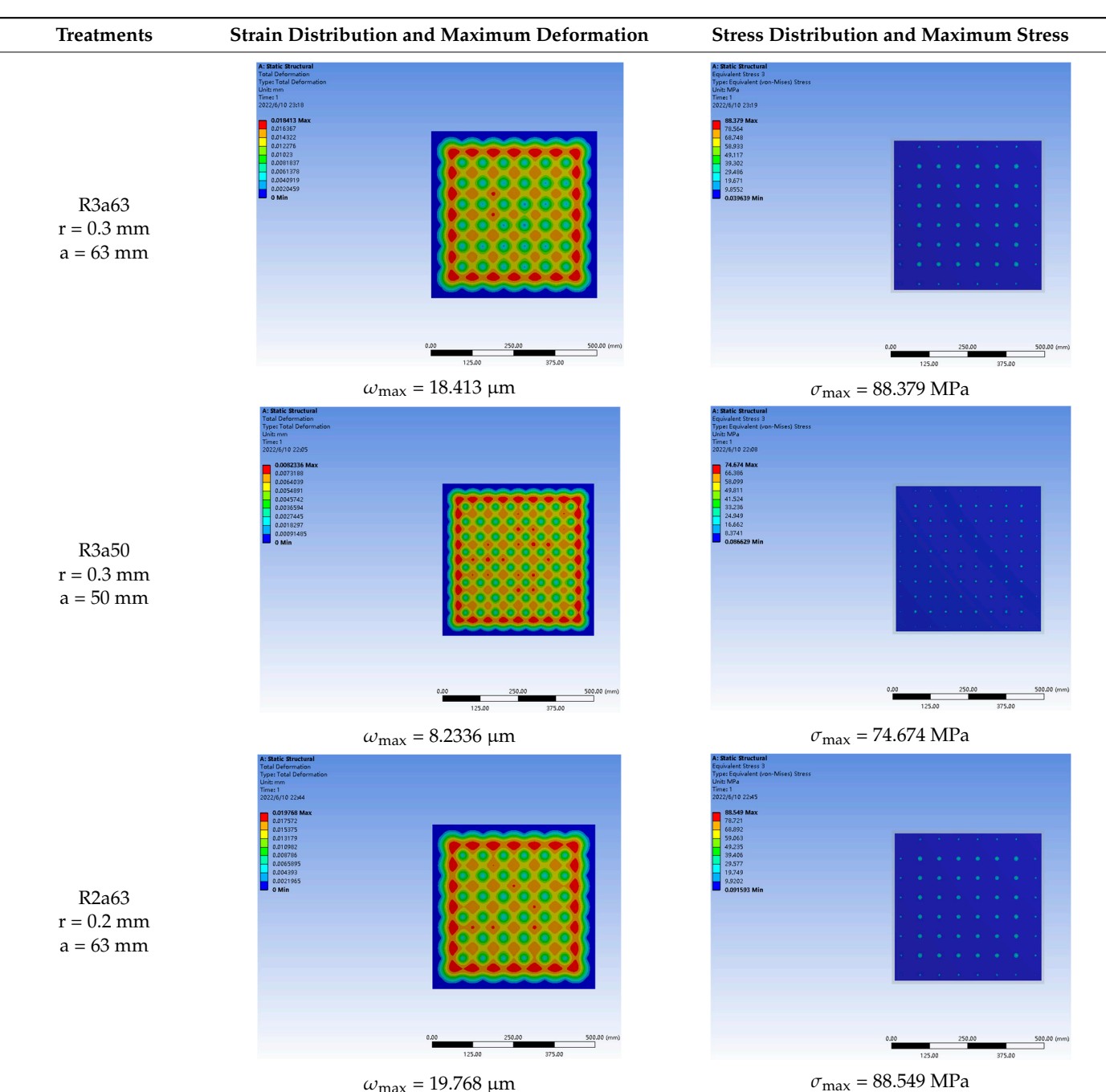

| Treatments | Strain Distribution and Maximum Deformation | Stress Distribution and Maximum Stress |
| --- | --- | --- |
| R3a63 <br> r = 0.3 mm <br> a = 63 mm | $\omega_{max}$ = 18.413 μm | $\sigma_{max}$ = 88.379 MPa |
| R3a50 <br> r = 0.3 mm <br> a = 50 mm | $\omega_{max}$ = 8.2336 μm | $\sigma_{max}$ = 74.674 MPa |
| R2a63 <br> r = 0.2 mm <br> a = 63 mm | $\omega_{max}$ = 19.768 μm | $\sigma_{max}$ = 88.549 MPa |

## 6. Discussion

Supporting pillar placement is the principal problem in VIG manufacturing because it affects the stress concentration distribution, mechanical properties and thereby the product life. In this study, comprehensive, multifaceted mechanical analysis was performed and a multi-objective solution for supporting pillar placement was established based on the stress and deformation distribution. We obtained critical parameters for the supporting pillar radius and spacing distance. The findings through case simulation and verification help us understand how the supporting pillar determines VIG mechanical properties.

Firstly, due to the quadrilateral placement of supporting pillars, the stress and deformation concentration points of the glass pane showed a quadrilateral array distribution tendency. The way to improve mechanical properties for VIG mainly focused on optimizing the maximum stress value in the stress concentration area.

Secondly, glass pane parameters, such as elastic modulus, thickness and Poisson ratio, inevitably affect the stress and deformation distribution. Therefore, it is necessary to design the critical supporting parameters to ensure the mechanical stability for VIG, according to the pane-specific structural and material properties. This paper provided a multi-objective solution to solve critical spacing parameters, so that we could obtain the acceptable scheme for the supporting pillar placement according to the design requirements.

However, this study has some limitations. Firstly, the feasibility of the multi-objective critical model was verified by computer simulation. In fact, the mechanical properties of VIG also need to be subjected to sample testing and microstructure comparison. Second, a more profound understanding and interpretation of the stress characteristics of VIG is necessary. The results of this study are not sufficient to explain the stress situation under other supporting pillar placement forms such as regular triangle and regular hexagon supports.

## 7. Conclusions

This study provided a multi-objective solution to determine critical supporting parameters for VIG. Case simulation verified the feasibility of this method with multidimensional constraint conditions. Thus, a 0.2 mm or more radius and a 63 mm or less spacing distance for the supporting pillars were recommended as acceptable placement methods according to the design requirements.

Future studies will achieve multidimensional analysis for the mechanical properties of VIG, in terms of supporting indentation, microstructure, etc. More importantly, an interdisciplinary explanation would help us elucidate the deeper relationship between VIG mechanical stability and supporting forms such as pillar supports and ball supports. Various array methods for supporting should continue to be explored, for instance, square, regular triangle and regular hexagon supports. Above all, we can design supporting pillar placement parameters that meet different strength requirements as needed. This research will act as a reference for future studies, promoting the in-depth development of VIG and exploration of high-strength safety products.

**Author Contributions:** Conceptualization, Y.Z.; methodology, L.H. and W.Y.; software, W.Y.; validation, Y.Z. and W.Y.; formal analysis, Y.Z.; investigation, X.X.; resources, R.Z.; data curation, X.X.; writing—original draft preparation, Y.Z.; writing—review and editing, W.Y.; visualization, X.X.; supervision, R.Z.; project administration, X.X.; funding acquisition, R.Z. All authors have read and agreed to the published version of the manuscript.

**Funding:** This research was funded by National Natural Science Foundation of China (Grants Nos. 52002349 and 51902284).

**Institutional Review Board Statement:** Not applicable.

**Informed Consent Statement:** Not applicable.

**Data Availability Statement:** Not applicable.

**Conflicts of Interest:** The authors declare no conflict of interest.

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
