# Peer review of "Multi-Objective Design for Critical Supporting Parameters of Vacuum-Insulated Glazing with a Case Study"

_applsci, doi:10.3390/app12157504_

Round 1
Reviewer 1 Report
1. English language and style should be revised. Few sentences are not understandable. Below you may find few of them:
- line 58: new sentence from the word Laser,
- line 68: "...existing case was compare to verify..." There should be written what kin of comparison has been performed.
- support pillars - nto better supporting pillars?
- Line 87,88. Please rewrite this sentence to be more understandable;
- - Line 109-112: form of the sentence not proper for the article. It should be modified.
- - Line 135 - Maybe instead of "borne" it is better to write "taken"?
- - Line 144 - "At the edge sealing part" instead portion.
- - Line 144-146 - Sentence started from "Considering..." is not understable. Please correct it.
- - Line 146-147 - It is better not to use the imperative mode.
- - Line 162 - Sentence "Under the continuous..." should be rewritten.
- - Line 164-165 - Maybe it would be better to write: "The maximum bending deformation in glass substrate is generated..."
- - It would be better to not repeat all the time "glass substrate" and to find synonym ( for example glass pane).
- - Make size of the symbols in the equations unified.
- - Line 194 -" (... , let L be plate dimension, S the number of support pillar in each row:") I think it is better this way.
- Line 25,26 – sounds like program tutorial. Please revise manuscript to delete imperative mode, especially point 5, 6 7.
- In Table 3 results are not visible. Please check again stress results and deformation results.
2. Consider to enrich your literature with more positions. Especially regarding pillar placement. Exemplary positions:
- https://doi.org/10.1016/j.conbuildmat.2021.125724
- https://doi.org/10.3390/ma15041467
- https://doi.org/10.5762/KAIS.2012.13.3.1002
3. Equation 11 - there is no information about the origin of the equation (literature position, is it a pariculat solution, how the equation was obtained)
4. For the equation 1 proper Figure is needed.
5. Equation 4 - Why there is umber 2 next to M1 and M2 repespectively?
6. Keep the same symbols in all equations. For example in equation 8 is different symbol for multiplication.
7. How did you get to the equation number 10 and 11. It should be described more, figure would be of great help.
8. You have assumed quite big simplifications for your model. Mainly: no boundary conditions; only part of the plate has been consider. You wrote in the line 159 that the first support pillar doesn't contact with the glass. Please explain that foundation. Pillars cannot stay without any connection.
9. Equation 12 - what does s.t. mean? You cannot use the same symbolic for different meaning (m was edge sealing width , here means something else, similar is with E). What does E mean in equation 12?
10. Equation in line 199 seems to be wrtitte incorrectly. Please check it. f1 and f2 have different meaning in the previous equations (number 1,2)
11. Line 209, Equation 17 - no explanation of the symbols.
12. I cannot find equality constraint function (h2).
13. Table 1: How the constant coefficients have been chosen ?
Reviewer 2 Report
The authors present a multi-objective optimization algorithm based on various stress mechanics on vacuum insulated glazing (VIG) along with results of a simulation case study. Spacing of the support pillars having various radius have been analyzed and compared for an optimal solution. Overall the authors make a fairly good attempt on an important aspect of mechanical stress of the VIG. Authors may cross check the title given for the first column in the Table1. The authors are suggested to cite some from references from MDPI journals.
Author Response
Please see the attachment.

This manuscript is a resubmission of an earlier submission. The following is a list of the peer review reports and author responses from that submission.